# TRAIN LONG, THINK SHORT: CURRICULUM LEARNING FOR EFFICIENT REASONING

## ABSTRACT

Recent work on enhancing the reasoning abilities of large language models (LLMs) has introduced explicit length control as a means of constraining computational cost while preserving accuracy. However, existing approaches rely on fixed-length training budgets, which do not take advantage of the natural progression from exploration to compression during learning. In this work, we propose a curriculum learning strategy for length-controlled reasoning using Group Relative Policy Optimization (GRPO). Our method starts with generous token budgets and gradually tightens them over training, encouraging models to first discover effective solution strategies and then distill them into more concise reasoning traces. We augment GRPO with a reward function that balances three signals: task correctness (via verifier feedback), length efficiency, and formatting adherence (via structural tags). Experiments on GSM8K, MATH500, SVAMP, College Math, and GSM+ demonstrate that curriculum-based training consistently outperforms fixed-budget baselines at the same final budget, achieving higher accuracy and significantly improved token efficiency. We further ablate the impact of reward weighting and decay schedule design, showing that progressive constraint serves as a powerful inductive bias for training efficient reasoning models.

## 1 INTRODUCTION

Recent advances in large language models (LLMs) have enabled impressive capabilities across a wide range of natural language processing tasks. A key challenge now is equipping these models with robust *reasoning* abilities, enabling them to solve problems that require systematic, multistep inference. To date, two main paradigms have emerged to improve reasoning in LLMs. The first relies on *supervised fine-tuning* (SFT) on datasets containing *chain-of-thought* (CoT) annotations, where human experts provide intermediate reasoning steps. Although SFT is straightforward to implement, it depends on costly data collection and may struggle to generalize beyond seen distributions. The second paradigm uses *reinforcement learning* (RL) to directly optimize the behavior of the model through feedback on the completed reasoning traces. RL-based methods avoid explicit reasoning annotations, can leverage sparse rewards, and have achieved state-of-the-art performance recently.

Within the RL category, *Group Relative Policy Optimization* (GRPO) has shown particular promise. GRPO fine-tunes LLMs without a separate value function by sampling a group of candidate responses per prompt and normalizing rewards across that group. This group-relative normalization stabilizes learning from sparse correctness signals and encourages the model to prefer responses that are strong relative to its own cohort.

An orthogonal line of work incorporates explicit *length control* into reasoning training: models are trained to produce reasoning traces under token-budget constraints, balancing solution quality and efficiency. Prior methods that handle multiple fixed budgets independently fail to leverage the natural progression of capability that can arise if the model is first allowed longer reasoning chains and then gradually required to compress them.

In this paper, we introduce curriculum learning for length-controlled reasoning. Instead of fixing the budget throughout the training, we begin with a large initial token budget $B_0$ and progressively tighten it via an exponential decay schedule:

$$B(t) = \max\left(1, \ B_0 \cdot \gamma^{\left\lfloor \frac{t}{T} \right\rfloor}\right),$$

where $\gamma \in (0, 1)$ is the decay factor and $T$ is the step interval between budget updates. During training, the model can explore a long chain-of-thought to discover effective reasoning patterns; as the budget shrinks, it is forced to distill these patterns into more concise and efficient reasoning traces.

We train with GRPO-based curriculum length control on two complementary mathematical reasoning datasets: GSM8K and MATH500. We then evaluate zero-shot performance on GSM8K, MATH500, SVAMP, College Math, and GSM+, comparing against fixed-budget GRPO baselines and base models without reasoning fine-tuning. Our experiments, conducted with QWEN-2.5-7B, show that curriculum learning yields consistent gains in both accuracy and token efficiency at the same final budget, indicating that progressive constraint is a powerful inductive signal for efficient reasoning.

Our contributions are as follows.

1. We propose a curriculum learning strategy for length-controlled reasoning by embedding an exponentially decaying token budget into GRPO fine-tuning, enabling a smooth transition from exploration to compression of reasoning chains.

2. We empirically demonstrate that curriculum-based length control outperforms fixed-budget training across multiple benchmarks, improving reasoning accuracy while reducing average token usage.

3. We release a reproducible implementation built on `torchtune` along with pretrained checkpoints to accelerate future work on LLMs capable of efficient reasoning.

## 2 RELATED WORK

**Test-Time Scaling and the Rise of Long-Chain Reasoning.** A dominant trend in enhancing the reasoning capabilities of LLMs is increasing computation at inference time. This strategy, often termed test-time scaling, has consistently improved performance in complex reasoning tasks, from mathematics to code generation (Wang et al., 2023; Wu et al., 2025a; Wei et al., 2022). Prominent approaches include sampling multiple reasoning paths and selecting the most consistent answer (self-consistency) (Wang et al., 2023), exploring solution paths with tree-based search (Yao et al., 2023), and iterative refinement (Madaan et al., 2023; Welleck et al., 2024). Recent state-of-the-art reasoning models, such as OpenAI's O1 and DeepSeek's R1-style models, are trained with reinforcement learning to generate extended reasoning traces, embodying a "think more" philosophy to tackle difficult problems (Jaech et al., 2024; Guo et al., 2025). However, this paradigm often leads to significant computational overhead and a phenomenon known as "overthinking," where models produce verbose and inefficient reasoning chains even for simple problems (Anonymous, 2025). These methods, while powerful, lack precise mechanisms to control the length of their outputs, creating a trade-off where higher accuracy comes at the cost of unpredictable and often excessive token usage.

**Approaches to Length Control and Reasoning Efficiency.** In response to the inefficiency of long-chain reasoning, a parallel line of research has focused on controlling the length of LLM outputs. Early work in this area addressed general text generation through architectural modifications (Butcher et al., 2025) or fine-tuning on instruction datasets labeled with desired lengths (Yuan et al., 2024). More recent work has tailored length control specifically for reasoning. Some approaches train models to generate shorter chains of thought (Arora & Zanette, 2025; Kang et al., 2025), while others use "budget-forcing" techniques that truncate outputs or pad with special tokens to meet a fixed limit (Muennighoff et al., 2025). However, these hard constraints can be suboptimal, as abrupt truncation can disrupt reasoning.

Other methods pursue finer-grained control by identifying and suppressing low-utility tokens at inference time. Xia et al. (2025) propose *TokenSkip*, a method that estimates the importance of the token and skips useless tokens to compress reasoning chains while preserving performance. Xu et al. (2025) present the *Chain of Draft* strategy, prompting models to write concise intermediate drafts rather than verbose step-by-step thoughts, dramatically reducing token usage without sacrificing accuracy. Wu et al. (2025b) provide a complementary analytical perspective, showing that accuracy follows an inverted-U curve with respect to chain length and proposing a length-aware voting heuristic that filters out traces that are too short or too long.

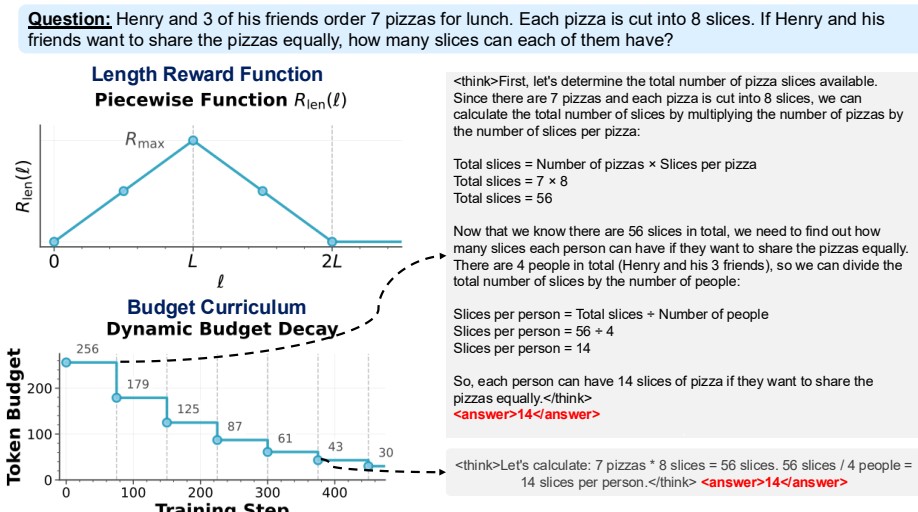

Figure 1: **Curriculum Learning GRPO Overview.** Our proposed setting performs GRPO with a length reward being applied to the generated thinking trace. The budget is decayed exponentially with a user specific decay factor and decay interval. In this example the decay factor $\gamma$ is set to $0.7$ and the decay interval $T$ is set to $100$. An initial budget of $256$ tokens is given at start and decayed later down to $30$. The figure demonstrates that the model learns to answer the same question with a way smaller token budget reaching the same solution.

Reinforcement learning has also been applied to dynamically optimize reasoning length. Fang et al. (2025) introduce *Thinkless*, a policy learning framework that trains models to decide *when to think*, selecting between short and long reasoning paths based on the difficulty of the problem. Similarly, Dumitru et al. (2025) propose *ConciseRL*, which rewards models for generating correct but concise reasoning by incorporating a learned conciseness score into the RL reward function. These methods demonstrate that length control can be learned in a context-sensitive and adaptive way, enabling models to use fewer tokens on easier problems and longer chains only when necessary.

A more sophisticated approach is taken by Aggarwal & Welleck (2025) with **Length Controlled Policy Optimization** (LCPO). Their method uses reinforcement learning to train a model to adhere to a user-specified length budget provided directly in the prompt. The reward function optimizes for both task correctness and adherence to the target length, producing a model, L1, that can flexibly trade off accuracy and computational cost at inference time. This allows a user to request reasoning of a specific length (e.g., 512, 1024, or 2048 tokens) and receive a response that respects that budget. These methods, while offering powerful inference-time flexibility, treat length as a user-controlled parameter for a pre-trained, versatile model.

**Positioning Our Work.** Our work, "Train Long, Think Short," introduces a novel perspective by framing efficient reasoning as a *curriculum learning* problem. In contrast to previous methods such as LCPO (Aggarwal & Welleck, 2025), which train a model to respond to *user-specified length budgets at inference time*, our work investigates the training dynamic itself as a mechanism for optimization. We propose a dynamic training strategy where the budget is not a user-controlled parameter. Instead, it starts with a generous token budget that lets the model freely **explore** long reasoning paths, and then monotonically **decays** this budget so the same policy learns to *compress* its successful strategies into a concise form. The result is a standalone model that targets a tight token budget and, in practice, stays within roughly 5% of that limit on average, achieving substantial cost savings without any runtime user hints or prompt overhead.

## 3 METHODOLOGY

We build on Group Relative Policy Optimization (GRPO) and introduce a curriculum for length-controlled reasoning, augmented with explicit formatting and correctness signals. Our training sig-

nal for each generated completion combines three components: (1) a *correctness* reward based on automated verification, (2) a *length* reward encouraging adherence to a (curriculum-decayed) token budget, and (3) a *formatting* reward enforcing structured reasoning and answer separation via special tags. We first review the math behind GRPO, then formalize the prompt, define each reward component with its weighting, describe the curriculum schedule, and finally give the full optimization objective with refinements.

## 3.1 GRPO PRELIMINARIES

Given a prompt $s$, the current (old) policy $\pi_{\theta_{\mathrm{old}}}$ is used to sample a group of $G$ responses $\{a_i\}_{i=1}^{G}$. Each response $a_i$ is assigned a scalar reward $r_i$ (defined below). Let the empirical mean and standard deviation over the group be

$$\mu = \frac{1}{G} \sum_{i=1}^{G} r_i, \qquad \sigma = \sqrt{\frac{1}{G} \sum_{i=1}^{G} (r_i - \mu)^2 + \epsilon_{\mathrm{stab}}},$$

where $\epsilon_{\mathrm{stab}} > 0$ is a small stabilizer to avoid division by zero. The group-relative advantage is

$$A_i = \frac{r_i - \mu}{\sigma}.$$

Define the probability ratio

$$r_i^{\mathrm{ratio}} = \frac{\pi_\theta(a_i \mid s)}{\pi_{\theta_{\mathrm{old}}}(a_i \mid s)}.$$

The clipped surrogate GRPO objective with reference regularization is as follows:

$$J_{\mathrm{GRPO}}(\theta) = \mathbb{E}_s \left[ \frac{1}{G} \sum_{i=1}^{G} \min\left( r_i^{\mathrm{ratio}} A_i, \ \mathrm{clip}(r_i^{\mathrm{ratio}}, 1 - \epsilon, 1 + \epsilon) A_i \right) \right] - \beta \, \mathrm{KL}(\pi_\theta \,\|\, \pi_{\mathrm{ref}}),$$

where $\epsilon > 0$ controls the clipping window and $\beta$ trades off deviation from $\pi_{\mathrm{ref}}$.

## 3.2 PROMPT STRUCTURE

To explicitly separate internal reasoning from the final answer and to enforce a fixed-length constraint, we prompt the model with the following instruction:

> **Prompt Template**
>
> A conversation between User and Assistant. The user asks a question, and the Assistant solves it. The assistant first thinks about the reasoning process in the mind and then provides the user with the answer. The reasoning process and answer are enclosed within `<think></think>` and `<answer></answer>` tags, respectively, i.e., `<think>reasoning process here</think> <answer>answer here</answer>`. IMPORTANT: You should use exactly {token_budget} tokens in your response. User: {question} Assistant:

The ideal model output takes the form:

$$\langle\texttt{<think>}\rangle \text{ chain-of-thought reasoning } \langle\texttt{</think>}\rangle \quad \langle\texttt{<answer>}\rangle \text{ final answer } \langle\texttt{</answer>}\rangle,$$

and the total token count is guided toward the budget via the curriculum and associated reward.

## 3.3 REWARD DECOMPOSITION AND WEIGHTING

For each sampled response $a_i$ of length $\ell_i$ (in tokens), we define the total scalar reward as a weighted sum of three components:

$$r_i = \lambda_c \cdot r_i^{\mathrm{correct}} + \lambda_\ell \cdot R_{\mathrm{len}}(\ell_i) + \lambda_f \cdot R_{\mathrm{fmt}}(a_i),$$

where $\lambda_c$, $\lambda_\ell$, $\lambda_f$ are nonnegative scalar weights controlling the relative importance of correctness, length adherence, and formatting, respectively. This makes explicit the experimental 'weights' (e.g., correctness vs. length vs. formatting) used in different settings.

**Correctness Reward.** Let $c_i \in \{0, 1\}$ be the indicator that the final answer (extracted from within $\langle\texttt{<answer>}\rangle$) passes the automated verifier (`math-verify`)—either exact numeric/symbolic match or a graded acceptance if extended. Then:

$$r_i^{\text{correct}} = R_{\text{cor}} \cdot c_i,$$

where $R_{\text{cor}} > 0$ is the base correctness reward. Optionally, if the verifier provides partial scores or confidence, $c_i$ can be softened to $[0, 1]$ and $r_i^{\text{correct}}$ adjusted accordingly.

**Length Reward.** Let the current target length be $L = B(t)$ (see next subsection). We define a triangular (piecewise linear) reward that encourages matching $L$ without encouraging trivial short or excessively long outputs:

$$R_{\text{len}}(\ell) = \begin{cases} R_{\max} \cdot \dfrac{\ell}{L} & \text{if } 0 \leq \ell \leq L, \\ R_{\max} \cdot \left(1 - \dfrac{\ell - L}{L}\right) & \text{if } L < \ell \leq 2L, \\ 0 & \text{if } \ell > 2L, \end{cases}$$

where $R_{\max} > 0$ is the maximum length reward at $\ell = L$. This shape (ramp-up, plateau at peak, ramp-down, hard cutoff) encourages the model to use the budget efficiently. In practice, we clip $\ell$ when computing the length if a generation exceeds $2L$ to avoid inflated computation; those responses receive zero for the length component.

**Formatting Reward.** Define indicators $\mathbb{I}_{\text{think}}$ and $\mathbb{I}_{\text{answer}}$ that equal 1 if the output contains well-formed, non-overlapping $\langle\texttt{<think>}\rangle/\langle\texttt{</think>}\rangle$ and $\langle\texttt{<answer>}\rangle/\langle\texttt{</answer>}\rangle$ spans, respectively, and zero otherwise. Then:

$$R_{\text{fmt}}(a) = \alpha_{\text{think}} \cdot \mathbb{I}_{\text{think}} + \alpha_{\text{answer}} \cdot \mathbb{I}_{\text{answer}},$$

with $\alpha_{\text{think}}, \alpha_{\text{answer}} > 0$ rewarding proper structural separation. This encourages the model to clearly expose its reasoning and final answer in the prescribed format.

### 3.4 CURRICULUM TOKEN BUDGET

We impose a curriculum on the allowable token budget so that it decays exponentially over training steps, enabling a natural transition from exploration (long, rich reasoning) to compression (concise reasoning under tight constraints). Starting from an initial budget $B_0$, the budget at training step $t$ is:

$$B(t) = \max\left(1, \; B_0 \cdot \gamma^{\left\lfloor \frac{t}{T} \right\rfloor}\right),$$

where $\gamma \in (0, 1)$ is the decay factor and $T$ is the interval (in steps) between budget updates. The target length $L$ in $R_{\text{len}}$ is set to $B(t)$, making the length reward progressively stricter as training progresses.

## 4 EXPERIMENTS

To evaluate the efficacy of our curriculum learning approach, we train models on math-reasoning data and measure accuracy, token efficiency, and robustness to training hyperparameters. Our experiments address six key questions:

Q1: Does curriculum learning improve reasoning performance compared to fixed-budget training when both finish at the same token budget?

Q2: Are the gains consistent across training datasets of different complexity (the easier GSM8K vs. the harder MATH500)?

Q3: How sensitive is performance to reward weighting, i.e., how do different correctness-versus-length reward weights affect the accuracy–efficiency tradeoff?

Q4: How does the shape of the decay schedule impact the final accuracy–efficiency tradeoff?

Q5: How does the choice of *length reward function* (triangular vs. flat-band) influence the balance between output compression and accuracy?

Q6: How does the *budget decay schedule type* (exponential vs. linear) affect final performance and efficiency across tasks of varying difficulty?

We answer questions Q1–Q3 in the main text. For Q4–Q6, which ablate specific curriculum design choices, we provide a concise summary in Section 4.4 and present the full experimental results in Appendix A.

## 4.1 SETUP

**Model.** We use QWEN-2.5-7B in all experiments, fine-tuned via GRPO using group size $G = 8$.

**Baselines.** We compare models trained using three different approaches:

1. **Base model**: the original QWEN-2.5-7B without further training; this isolates the benefit of any budget-aware RL fine-tuning.

2. **Fixed-budget GRPO**: the same model fine-tuned with GRPO while enforcing a constant 87-token limit; this matches the final budget but isolates our proposed curriculum.

3. **Our Curriculum GRPO**: GRPO training with an exponential budget schedule that decays from 256 to 87 tokens.

**Training Data.** For each baseline, we train two checkpoints. One uses all 7,473 GSM8K grade-school problems, whose solutions are usually concise. The other uses MATH500, which represents 500 hard competition-level problems from the MATH dataset; these questions typically require longer chains of reasoning.

**Budget range.** We start the curriculum at 256 tokens, which is more than sufficient to solve most GSM8K problems and only just sufficient for many MATH500 problems. We then decay it exponentially to 87 tokens. This schedule tests whether gradual tightening compresses the chain-of-thought without reducing accuracy.

**Evaluation Datasets.** We evaluate zero-shot on five benchmarks: GSM8K (grade-school arithmetic), SVAMP (perturbed variants of GSM8K problems), and GSM+ (adversarial GSM8K problems), as well as MATH500 (competition-level math) and College Math (university-level math).

## 4.2 CURRICULUM LEARNING VS. FIXED BUDGET

We first test whether curriculum learning yields better token efficiency than the base model and higher accuracy than fixed-budget GRPO, in a setting where the curriculum and fixed-budget models finish training with the same 87-token limit. We train on either GSM8K (Figure 2 Top) or MATH500 (Figure 2 Bottom), and evaluate on both in-distribution datasets and out-of-distribution benchmarks. Across both training datasets and all evaluation benchmarks, curriculum learning improves accuracy while matching the token efficiency of fixed-budget GRPO at the same final budget and significantly reducing token usage relative to the base model.

**GSM8K-trained models.** As shown in Figure 2 top, when trained on GSM8K, curriculum learning improves ID accuracy from 82.71% (fixed-budget GRPO) to 86.20%, with nearly identical average token usage (88.8 vs. 87.0). In comparison, the base model uses 258.4 tokens to reach only 83.55% accuracy, highlighting both the accuracy and efficiency benefits of curriculum training. For OOD evaluation on datasets derived from GSM8K, curriculum learning boosts accuracy from 77.67% to 85.00% on SVAMP (perturbed word problems) and from 62.75% to 67.58% on GSM+ (adversarial variants), again with token counts closely matching the fixed-budget baseline.

**MATH500-trained models.** On the harder MATH500 dataset (Figure 2 bottom), curriculum learning raises accuracy from 38.80% (fixed-budget) to 43.40% while compressing average reasoning length from 179.3 to 137.1 tokens. This shows that the model can shorten even long-form solutions without sacrificing correctness. Similar to GSM8K training, we observe some OOD gains here too.

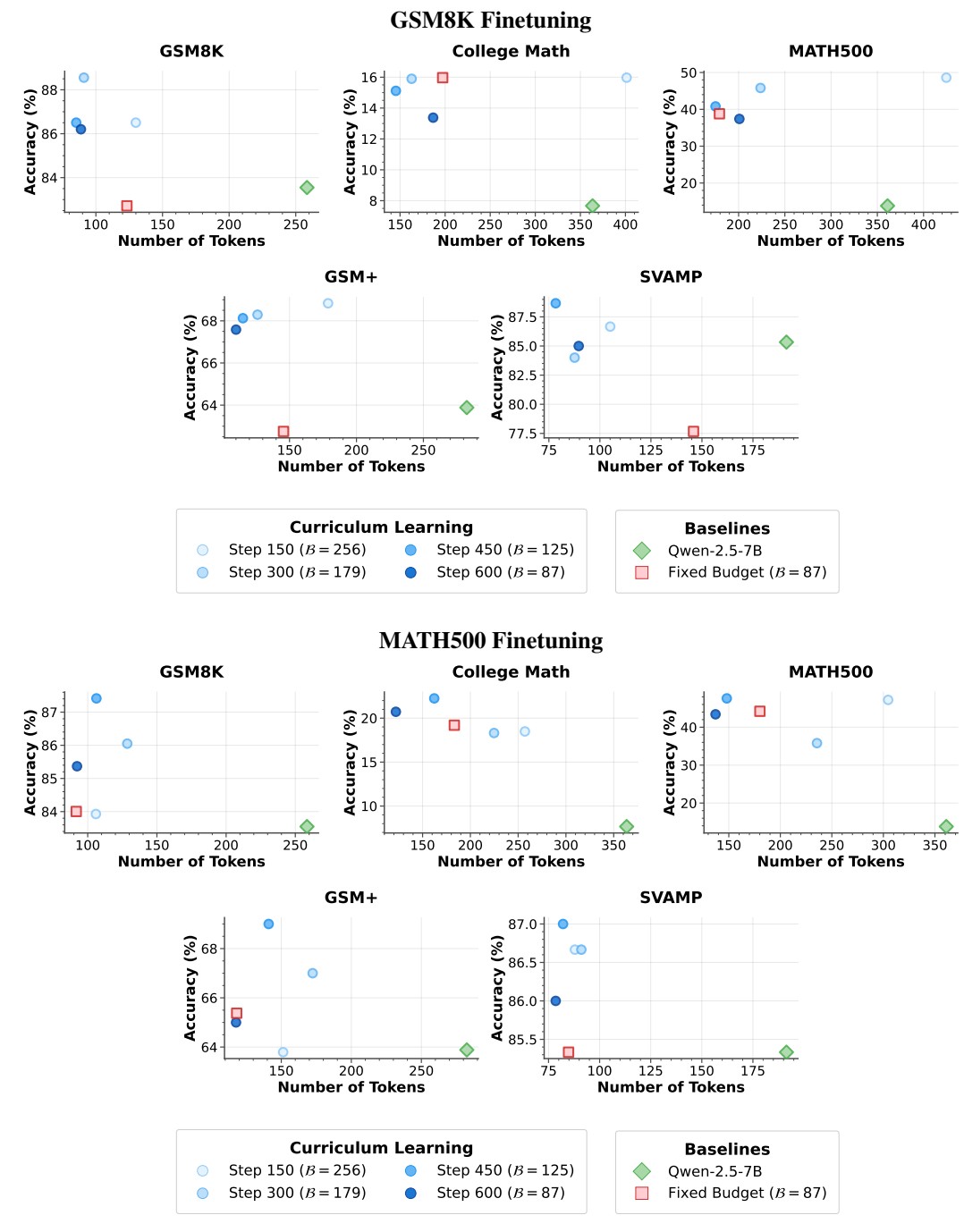

Figure 2: **Curriculum vs. fixed-budget training on GSM8K and MATH500.** For GSM8K (top), models trained with our curriculum (256 → 87 tokens) achieve higher in-distribution accuracy than fixed-budget GRPO at the same final budget, while using fewer tokens. For MATH500 (bottom), even for harder, longer-form problems, curriculum learning improves accuracy while reducing average reasoning length, showing that progressive budget tightening can compress solutions while maintaining high accuracy.

**Conclusion for Q1 & Q2.** In both easy (GSM8K) and hard (MATH500) reasoning tasks, curriculum learning consistently outperforms fixed-budget training in accuracy, while maintaining its token efficiency. In addition, it generalizes better to related perturbed or adversarial benchmarks.

## 4.3 REWARD WEIGHT ABLATIONS: CORRECTNESS VS. LENGTH

We next explore how varying reward weights impacts the tradeoff between solution quality and length. Figures 3 and 4 show two regimes: one prioritizing length ($\lambda_c = 0.3$, $\lambda_\ell = 0.6$) and one prioritizing correctness ($\lambda_c = 0.6$, $\lambda_\ell = 0.3$).

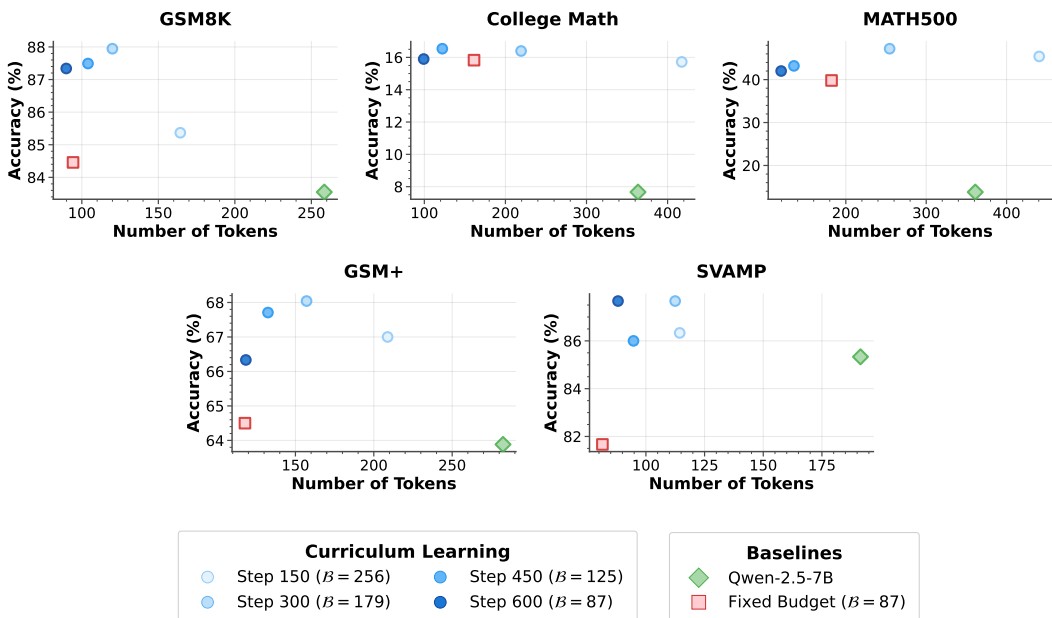

Figure 3: **Length-heavy reward weighting.** Increasing the weight on the length reward ($\lambda_c = 0.3$, $\lambda_\ell = 0.6$) yields highly compressed reasoning traces while retaining accuracy gains over the base model.

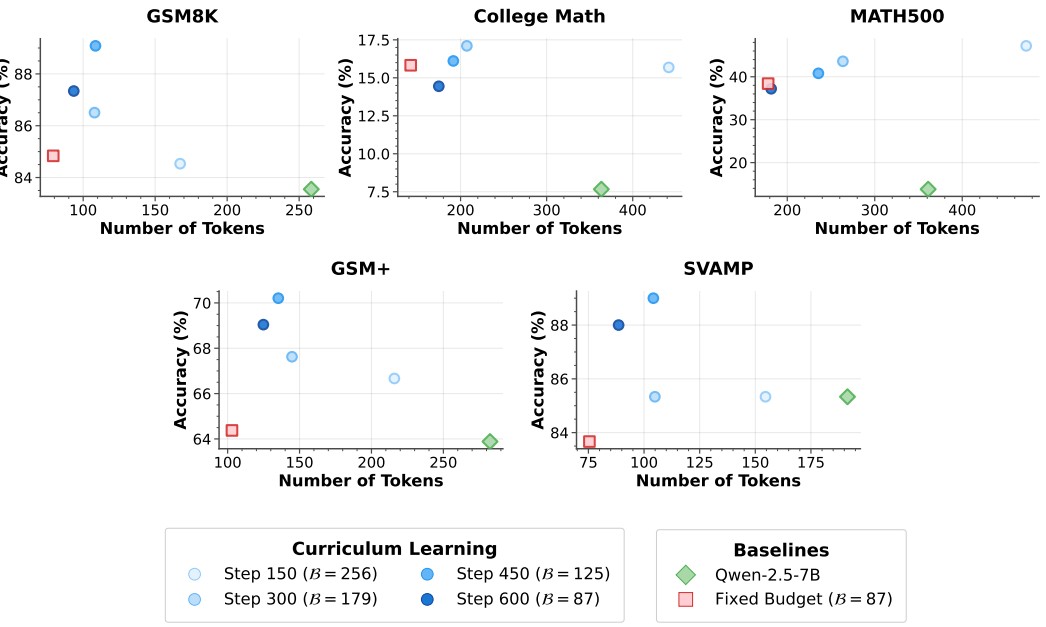

Figure 4: **Correctness-heavy reward weighting.** Prioritizing correctness ($\lambda_c = 0.6$, $\lambda_\ell = 0.3$) produces slightly longer outputs than the length-heavy setting but improves accuracy on both in-distribution and out-of-distribution benchmarks.

**Length-Heavy Setting (Figure 3).** At 600 steps (final budget), GSM8K accuracy reaches **85.37%** with an average length of **92.3** tokens, compared to the base model's **83.55%** at **258.4** tokens. This shows that emphasizing the length reward produces highly compressed reasoning traces while maintaining accuracy gains over the base.

**Correctness-Heavy Setting (Figure 4).** Shifting emphasis toward correctness improves GSM8K accuracy to **87.34%**, with a modest increase in average length to **93.5** tokens, still far below the base model. On SVAMP and GSM+, correctness-heavy training consistently outperforms the length-heavy setting by 1–2 points, confirming that higher accuracy comes at a small token cost.

**Conclusion for Q3.** Adjusting reward weights provides a controllable mechanism to trade accuracy for efficiency: heavier length weighting yields more compressed outputs with a slight accuracy drop, while heavier correctness weighting maximizes accuracy at a marginal increase in tokens.

### 4.4 ABLATION OF CURRICULUM DESIGN CHOICES

We briefly summarize our findings on how curriculum design choices affect performance. Full results and methodology for these experiments are available in Appendix A.

**Curriculum Schedule (Q4).** The trajectory of budget decay matters, not just the start and end points. Faster, more frequent budget reductions (Appendix A.1) tend to yield the best balance of accuracy and token efficiency, especially on harder tasks where a very slow decay can harm performance.

**Length Reward Function (Q5).** A *triangular* length reward, which encourages exploration up to the target budget before penalizing longer outputs, consistently outperforms a *band* reward (Appendix A.2). The band reward, which gives maximum points for any output below the budget, tends to over-compress solutions and harm accuracy, particularly on complex tasks.

**Decay Schedule Shape (Q6).** A *linear* decay schedule, which reduces the budget by a fixed amount at each step, achieves higher average accuracy than an *exponential* decay, especially on harder datasets like MATH500 (Appendix A.3). While exponential decay is slightly more token-efficient, its aggressive early compression can prematurely remove reasoning capacity. Linear decay offers a steadier compression path, better preserving performance on complex problems.

## 5 CONCLUSION

We introduced a curriculum learning framework for efficient reasoning in large language models, where token budgets decay over training time rather than remain fixed. Based on Group Relative Policy Optimization (GRPO), our approach combines three reward signals: correctness, length efficiency, and formatting structure to guide learning under progressively tighter constraints.

Our experiments show that curriculum-based training consistently improves both accuracy and token usage over fixed-budget baselines across multiple reasoning benchmarks. These gains hold whether training on simple arithmetic tasks (GSM8K) or competition-level mathematics (MATH500), and extend to adversarial and out-of-distribution evaluations.

We also show that the shape of the curriculum, that is, the rate and schedule of budget decay, significantly affects the final performance. Smoother decay paths encourage better compression without hurting accuracy, particularly on tasks requiring deeper reasoning. Finally, we show that reward composition (correctness vs. length emphasis) enables controllable trade-offs between solution quality and inference cost.

Together, these results suggest that curriculum-driven compression is a powerful and generalizable approach for training efficient reasoning models. We hope our open-source implementation and findings serve as a foundation for future work on budget-aware, verifiable, and scalable language model reasoning.

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

# A DETAILED ABLATION STUDIES ON CURRICULUM DESIGN

## A.1 EFFECT OF CURRICULUM SCHEDULE (Q4)

We now turn to investigate how the *shape* of the curriculum—i.e., the rate at which the token budget decays—impacts final performance. While all schedules begin at the same initial budget ($S_0 = 256$) and end at the same final budget ($S_f = 87$), we vary the number of decay points $n$, which determines how rapidly or gradually the model is constrained.

**Step-wise Exponential Schedule.** To ensure flexibility and principled control, we define a budget schedule updated every $I = T/(n + 1)$ steps, where $T$ is the total number of training steps. The budget after decay index $k$ ($k = 0, \ldots, n$) is

$$S_k = S_0 \cdot d^k, \quad \text{with} \quad d = \left(\frac{S_f}{S_0}\right)^{1/n},$$

applied at step $t_k = k \cdot I$. This ensures all schedules reach $S_f$ at the same endpoint while varying the decay trajectory. For example:

- $n = 1$, $d \approx 0.340$: single large, abrupt decay halfway through training.
- $n = 3$, $d \approx 0.700$: moderate decay every 150 steps ($T = 600$).
- $n = 7$, $d \approx 0.857$: gentle, gradual decay every 75 steps.

**Results.** Table 1 shows that decay trajectory substantially influences the final accuracy–efficiency trade-off, even with identical start and end budgets. On average across all datasets, fast ($I = 75$) and moderate ($I = 150$) decays achieve the highest mean accuracy (**57.9%**) while keeping token usage low (**115** and **135** tokens, respectively). Slow decay ($I = 300$) maintains higher token counts (**248** average) and matches or slightly exceeds the best accuracies on easier datasets like GSM8K (**86.8%**) and SVAMP (**88.0%**), but is far less efficient, and is less performant on hard datasets. A notable example is MATH500, where slow decay yields only **9.8%** accuracy, suggesting that very late decay harms performance on harder, long-form reasoning tasks.

Table 1: **Decay rate ablation (exponential schedules).** Fast and moderate decays deliver the highest average accuracy at substantially lower token budgets, while slow decay attains the best results on easier datasets (GSM8K, SVAMP) but performs poorly on harder tasks (MATH500) and is least efficient; start and end budgets are fixed across settings.

| Dataset | Decay | Interval | Avg. Token Count | Accuracy (%) |
|---|---|---|---|---|
| | 0.340 | 300 | 178 | **86.8** |
| **GSM8K** | 0.700 | 150 | 89 | 86.2 |
| | 0.857 | 75 | 103 | 84.7 |
| | 0.340 | 300 | 357 | 10.1 |
| **College Math** | 0.700 | 150 | 187 | 13.4 |
| | 0.857 | 75 | 119 | **15.3** |
| | 0.340 | 300 | 204 | 67.5 |
| **GSM+** | 0.700 | 150 | 110 | **67.6** |
| | 0.857 | 75 | 124 | 66.6 |
| | 0.340 | 300 | 167 | **88.0** |
| **SVAMP** | 0.700 | 150 | 90 | 85.0 |
| | 0.857 | 75 | 96 | 84.3 |
| | 0.340 | 300 | 336 | 9.8 |
| **MATH500** | 0.700 | 150 | 201 | 37.4 |
| | 0.857 | 75 | 132 | **38.4** |
| | 0.340 | 300 | 248 | 52.4 |
| **Average** | 0.700 | 150 | 135 | **57.9** |
| | 0.857 | **75** | **115** | **57.9** |

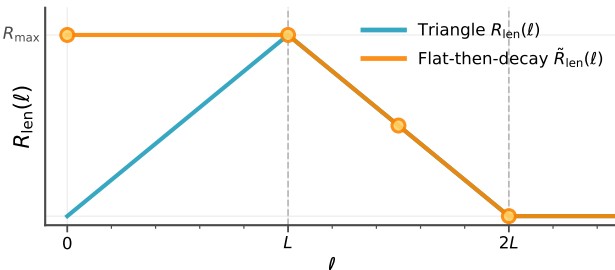

Figure 5: **Triangular vs. Band length reward.** The triangular shape encourages exploration up to the budget $L$ before compression, whereas the band shape gives maximum reward immediately for any output $\leq L$, often leading to shorter but less accurate reasoning traces. We refer to 'band' here as 'flat-then-decay'.

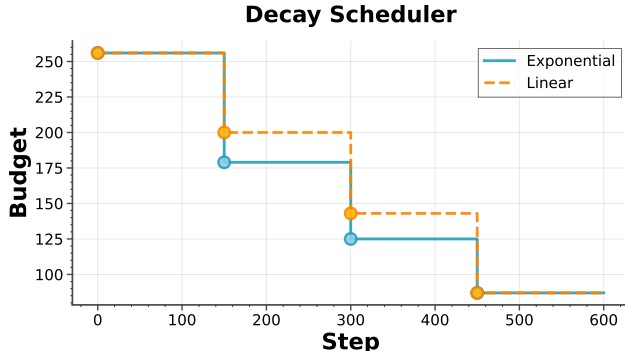

Figure 6: **Exponential vs. Linear decay schedules.** Linear decay reduces the budget in equal steps, leading to slightly longer outputs but improved performance on harder reasoning tasks.

**Conclusion for Q4.** The curriculum trajectory, not just the endpoint, matters. Faster decays favor efficiency and robustness on challenging tasks, while slower decays allow more exploration early, benefiting easier datasets. Our step-wise exponential framework provides a single tunable parameter $n$ to control this trade-off.

## A.2 EFFECT OF LENGTH REWARD FUNCTION (Q5)

In our main experiments, the length component of the reward function is implemented as a *triangular* shape (Section 3), which linearly increases from 0 at length 0 to a maximum at the target budget ($L = 87$), then linearly decreases to 0 at $2L$. This structure encourages the model to explore the full budgeted reasoning length, since using tokens up to $L$ yields progressively higher reward.

As an alternative, we evaluate a *band* reward function, where the length reward remains at a fixed maximum for all outputs up to $L$ tokens, and then decreases linearly to 0 at $2L$. This variant removes the ramp-up phase and gives maximal reward even for very short completions, which may encourage the model to settle on shorter-than-necessary reasoning traces if they already solve the task correctly.

**Results.** Figure 5 and Table 2 summarize the comparison. Across all datasets, we observe a clear trade-off: the band reward consistently produces shorter outputs (average 94 tokens vs. 135), but the triangular reward always achieves higher accuracy. The accuracy drop is especially noticeable on hard datasets such as MATH500 (30.8% vs. 37.4%) and GSM+ (64.6% vs. 67.6%). The triangular reward, by contrast, preserves accuracy while still achieving large efficiency gains over the base model, suggesting that incentivizing gradual length exploration before compression is beneficial.

Table 2: **Length reward shape comparison.** Triangular rewards encourage full-budget exploration before compression, yielding higher accuracy at similar efficiency, whereas band rewards often over-compress and lose performance.

| Dataset | Reward Function | Avg. Token Count | Accuracy (%) |
|---|---|---|---|
| GSM8K | Triangular | 89 | **86.2** |
| | Band | 70 | 84.6 |
| College Math | Triangular | 187 | **13.4** |
| | Band | 132 | 13.1 |
| GSM+ | Triangular | 110 | **67.6** |
| | Band | 98 | 64.6 |
| SVAMP | Triangular | 90 | **85.0** |
| | Band | 61 | 82.0 |
| MATH500 | Triangular | 201 | **37.4** |
| | Band | 112 | 30.8 |
| Average | Triangular | 135 | **57.9** |
| | Band | **94** | 55.0 |

Table 3: **Decay scheduler type comparison.** Exponential decay favors efficiency by front-loading compression, while linear decay provides steadier budget reduction, often improving performance on complex reasoning tasks.

| Dataset | Decay Scheduler | Avg. Token Count | Accuracy (%) |
|---|---|---|---|
| GSM8K | Exponential | 89 | 86.2 |
| | Linear | 107 | **86.3** |
| College Math | Exponential | 187 | 13.4 |
| | Linear | 154 | **17.2** |
| GSM+ | Exponential | 110 | **67.6** |
| | Linear | 143 | 66.4 |
| SVAMP | Exponential | 90 | 85.0 |
| | Linear | 97 | **87.3** |
| MATH500 | Exponential | 201 | 37.4 |
| | Linear | 198 | **42.8** |
| Average | Exponential | **135** | 57.9 |
| | Linear | 140 | **60.0** |

**Conclusion for Q5.** The triangular reward balances exploration and compression, achieving higher accuracy at similar efficiency to the band reward, which tends to over-compress and harm performance on harder tasks requiring longer-form reasoning (e.g., $-6.6$ points on MATH500).

A.3   EFFECT OF DECAY SCHEDULE SHAPE (Q6)

In addition to the reward shape, the *schedule* by which we decay the token budget may influence learning dynamics. Our default setting uses an *exponential* decay, where the budget is multiplied by a constant factor at fixed intervals (e.g., every 150 steps) until the final target length is reached. This produces a steep budget drop early on and increasingly smaller changes later.

As a comparison, we experiment with a *linear* decay schedule that reduces the budget in equal steps from the initial 256 tokens to the final 87 over the same total training duration. In our implementation, we perform roughly three equal budget drops to cover this range.

Figure 6 and Table 3 report the results. The linear schedule generally yields slightly longer outputs (average 140 tokens vs. 135 for exponential), but improves average accuracy from 57.9% to 60.0%. Gains are most pronounced on harder datasets like MATH500 (42.8% vs. 37.4%) and College Math

(17.2% vs. 13.4%), suggesting that a gentler, more uniform reduction in budget may help models retain complex reasoning strategies while still learning to compress them.

**Conclusion for Q6.** While exponential decay favors shorter outputs and slightly better average efficiency, it can remove reasoning capacity too quickly. In contrast, linear decay provides a steadier compression trajectory, yielding notable accuracy improvements on complex reasoning tasks.

## B   LIMITATIONS

Our study is limited in several respects due to computational constraints. First, all training was conducted with relatively short context windows and token budgets capped at 256 tokens. Although this suffices for datasets like GSM8K, it may restrict performance on tasks that require more extended reasoning. Extending curriculum learning to larger context windows could yield further gains.

Second, we conduct all experiments using the QWEN-2.5-7B model. While this model size provides a strong trade-off between capability and cost, it remains an open question how curriculum-based length control behaves at both larger (e.g., 13B, 70B) and smaller (e.g., 1.3B, 3B) scales. Scaling analyses and evaluations on open-ended generation tasks are promising directions for future work.

## C   USE OF LANGUAGE MODELS IN WRITING

We used large language models (LLMs) (ChatGPT, Gemini and Claude) to assist in drafting and refining the text of this paper. These tools supported clarity, coherence, and stylistic consistency throughout the writing process.

