# OpenReview forum: "Train Long, Think Short: Curriculum Learning for Efficient Reasoning"
_ICLR.cc/2026/Conference — Submitted to ICLR 2026_

### Official Review · Reviewer_8P1W · 2025-10-27

**Soundness:** 2
**Presentation:** 3
**Contribution:** 2
**Rating:** 4
**Confidence:** 4

**Summary:**

This paper presents a GRPO-based curriculum learning approach for length-controlled reasoning. The method begins with a generous token budget and progressively decays it during reinforcement learning fine-tuning. The authors design a composite reward that integrates accuracy, length efficiency, and formatting quality. Experimental results show that this curriculum-based training consistently outperforms fixed-budget baselines under the same final token budget, achieving higher accuracy and markedly improved token efficiency.

**Strengths:**

1. Treating length control as a curriculum rather than a fixed constraint is simple and compelling.

2.  Experimental results demonstrate that their method improves efficiency while maintaining and even improving the accuracy.

**Weaknesses:**

1. The use of curriculum learning with GRPO is not novel [1]. Moreover, the design of the length reward function closely resembles that in [2]. The authors should provide a more comprehensive discussion of these related works to clarify the differences and highlight their contributions.

2. Although the authors mention their limited computational resources, the generation length used in the experiments is too short (starting at only 256 tokens). As shown in Figure 1, the model’s reasoning traces resemble direct step-by-step solutions—similar to those produced by the original model—rather than genuine “thinking” processes. Moreover, Qwen2.5-7B is primarily an instruction-tuned model rather than a reasoning model. The authors should evaluate their method on dedicated reasoning models such as DeepScaleR-1.5B, Qwen3-1.5B, or DeepSeek-Distilled-1.5B to better demonstrate the generality of their approach.


[1] Hou, Bairu, et al. "Thinkprune: Pruning long chain-of-thought of llms via reinforcement learning." arXiv preprint arXiv:2504.01296 (2025).

[2] Huang, Chengyu, Zhengxin Zhang, and Claire Cardie. "HAPO: Training Language Models to Reason Concisely via History-Aware Policy Optimization." arXiv preprint arXiv:2505.11225 (2025).

**Questions:**

See Weaknesses.

---

### Official Review · Reviewer_9CbU · 2025-10-28

**Soundness:** 2
**Presentation:** 2
**Contribution:** 1
**Rating:** 2
**Confidence:** 3

**Summary:**

This paper proposes a curriculum learning for efficient reasoning. Starting from 256 tokens, the model is trained to gradually use smaller budget to preserve better accuracy.

**Strengths:**

- This paper proposes a simple curriculum method for enabling efficient reasoning.
- The paper is easy to read and follow.
- Extensive results show that at least it is better than using a fixed budget with a shorter budget.

**Weaknesses:**

- The training loss is just the same as previous papers, and the only difference is the curriculum. But is it really a more effective strategy compared to having a fixed-budget optimization like LCPO? It is highly skeptical. Novelty is also an issue here.
- Another following problem is the baselines. The baselines are very weak. It does not compare with performance fine-tuning like pure GRPO, so we don’t know how it trades off performance and efficiency. Also, a lot of efficient fine-tuning methods (LCPO, Adaptive Length Penalty, etc) should be included as baselines.
- Scheduling and the number of tokens are also unusual. The final token length, which is less than 100, is kind of impractical and not aligned with the philosophy of removing overthinking. Also, are there any experiments with increasing token budgets? That’s probably the reason why it is not good at hard benchmarks.
- Qwen7B-only experiments also weaken the paper.

**Questions:**

The training is applied to base models. I wonder whether just further training on instruction-tuned models is more efficient and effective.

---

### Official Review · Reviewer_KYeZ · 2025-10-29

**Soundness:** 2
**Presentation:** 3
**Contribution:** 2
**Rating:** 4
**Confidence:** 3

**Summary:**

This paper proposes a curriculum learning strategy for length-controlled reasoning. Our method gradually tightens the token budgets over training. So the method could first encourage models to discover effective solution strategies and then distill them into more concise reasoning traces. The method designs a new reward function that balances three signals: task correctness (via verifier feedback), length efficiency, and formatting adherence (via structural tags). The paper conduct experiments on 5 different datasets and show clear improvement in token efficiency.

**Strengths:**

1. The experiments address six key questions, which are helpful in understanding the improvement.
2. The paper focuses on an important question.
3. The paper proposes a simple but useful method.

**Weaknesses:**

1. The experiments are only conducted with QWEN-2.5-7B on math reasoning tasks. I think it will be helpful to show results on different model sizes and model families.
2. The user still needs to set a token budget, but it's hard for the users to know what the ideal budget is for each dataset. For example, hard questions need more tokens while easy questions need fewer.
3. I think some important baselines are missing. [1] Also target getting the best trade-off with different lengths. [1,2] train different budgets at the same time, while this paper trains different budgets with curriculum learning. There is not enough evidence to show that curriculum learning is better.
4. Recent papers also add a length reward to control the length, such as [1,2]. They also first train with a longer length than compressed, which is similar to the curriculum learning in this paper. So I think the novelty of the paper is a it limited.

[1] Zhang, Xuechen, et al. "Making small language models efficient reasoners: Intervention, supervision, reinforcement." arXiv preprint arXiv:2505.07961 (2025).

[2] Aggarwal, Pranjal, and Sean Welleck. "L1: Controlling how long a reasoning model thinks with reinforcement learning." arXiv preprint arXiv:2503.04697 (2025).

**Questions:**

See weakness

---

### Official Review · Reviewer_Ye3N · 2025-11-10

**Soundness:** 3
**Presentation:** 2
**Contribution:** 2
**Rating:** 4
**Confidence:** 3

**Summary:**

This paper introduces a curriculum learning approach for training efficient reasoning models, where instead of maintaining fixed token budgets throughout training, the authors start with generous budgets (256 tokens) that exponentially decay to tight constraints (87 tokens) over time. Built on GRPO, their method combines three reward signals: task correctness via automated verification, length efficiency, and formatting adherence using structured tags to guide the model through a natural progression from exploring long reasoning chains to compressing them into concise forms. Experiments on mathematical reasoning benchmarks using QWEN-2.5-7B demonstrate that curriculum-based training consistently achieves higher accuracy and better token efficiency than fixed-budget baselines at equivalent final budgets, with extensive ablations showing that the decay trajectory, reward shape, and schedule type all meaningfully impact the accuracy-efficiency tradeoff.

**Strengths:**

The paper presents a novel perspective on efficient reasoning training by framing it as a curriculum learning problem, which aligns intuitively with how models should progress from exploration to compression. The experimental evaluation is thorough and systematic, with consistent improvements across multiple benchmarks and comprehensive ablation studies examining reward weighting, decay schedules, reward shapes, and decay intervals. The clear visualization of the approach and detailed methodology make the work reproducible, and the authors' commitment to releasing implementation and checkpoints adds practical value to the community.

**Weaknesses:**

The study is significantly limited by its scope. All experiments use only QWEN-2.5-7B, leaving open questions about scaling behavior across model sizes, and the token budget is capped at 256, which may not reflect realistic reasoning scenarios requiring longer chains. The training datasets are relatively modest, and while improvements are consistent, they are sometimes modest, raising questions about practical significance. Additionally, the method introduces multiple hyperparameters requiring tuning, and the formatting tag requirement may not be practical for all applications; the paper also provides limited empirical comparison to other recent length-control methods beyond positioning relative to LCPO.

**Questions:**

How does curriculum learning scale to larger models (13B, 70B) and longer token budgets, and does the benefit persist or diminish? Does this approach work effectively on reasoning tasks beyond mathematics, such as code generation, logical reasoning, or open-ended problem-solving? What is the actual computational overhead of curriculum training compared to fixed-budget training, and does the improved final efficiency justify any training-time costs? How sensitive is performance to the specific choice of decay factor and interval, and can these be automatically tuned? Can curriculum learning be combined synergistically with other efficiency techniques like test-time scaling, pruning, or distillation? Finally, what is the role of the verifier quality, and how robust is the approach to imperfect or biased correctness signals?

---

### Meta-Review · Area_Chair_mzYX · 2025-12-15

**Summary:**

The paper proposes a curriculum learning strategy for learning length-controlled reasoning. The reviewers did not find the method sufficiently compelling and there is no rebuttal from the authors.

**Reviewer Concerns:**

The reviewers mostly did not find enough novelty in the work. There are also concerns that the work only used one model.

**Reviewer Scores:**

N.A.

---

### Decision · Program_Chairs · 2026-01-26

Reject